# Anti-Vascular Inflammatory Effect of Ethanol Extract from *Securinega suffruticosa* in Human Umbilical Vein Endothelial Cells

**DOI:** 10.3390/nu12113448

**Published:** 2020-11-10

**Authors:** Byung Hyuk Han, Chun Ho Song, Jung Joo Yoon, Hye Yoom Kim, Chang Seob Seo, Dae Gill Kang, Yun Jung Lee, Ho Sub Lee

**Affiliations:** 1Hanbang Cardio-Renal Syndrome Research Center, Wonkwang University, 460, Iksan-daero, Iksan 54538, Jeonbuk, Korea; arum0924@naver.com (B.H.H.); schsongaz@naver.com (C.H.S.); mora16@naver.com (J.J.Y.); hyeyoomc@naver.com (H.Y.K.); dgkang@wku.ac.kr (D.G.K.); 2College of Oriental Medicine and Professional Graduate School of Oriental Medicine, Wonkwang University, 460, Iksan-daero, Iksan 54538, Jeonbuk, Korea; 3Herbal Medicine Research Division, Korea Institute of Oriental Medicine, 1672 Yuseong-daero, Yuseong-gu, Daejeon 34054, Korea; csseo0914@kiom.re.kr

**Keywords:** vascular inflammation, *Securiniga suffruticosa*, HUVECs, TNF-α, adhesion molecules, NO

## Abstract

*Securiniga suffruticosa* is known as a drug that has the effect of improving the blood circulation and relaxing muscles and tendons, thereby protects and strengthen kidney and spleen. Therefore, in this study, treatment of *Securiniga suffruticosa* showed protective effect of inhibiting the vascular inflammation in human umbilical vein endothelial cells (HUVECs) by inducing nitric oxide (NO) production and endothelial nitric oxide synthase (eNOS) coupling pathway. In this study, *Securiniga suffruticosa* suppressed TNF-α (Tumor necrosis factor–α) induced protein and mRNA levels of cell adhesion molecules such as intracellular adhesion molecule-1 (ICAM-1), vascular cell adhesion molecule-1 (VCAM-1) and Interleukin-6 (IL-6). Pretreatment of HUVEC with *Securiniga suffruticosa* decreased the adhesion of HL-60 cells to Ox-LDL (Oxidized Low-Density-Lipoprotein)-induced HUVEC. Moreover, *Securiniga suffruticosa* inhibited TNF-α induced intracellular reactive oxygen species (ROS) production. *Securiniga suffruticosa* also inhibited phosphorylation of IκB-α in cytoplasm and translocation of NF-κB (Nuclear factor-kappa B) p65 to the nucleus. *Securiniga suffruticosa* increased NO production, as well increased the phosphorylation of eNOS and Akt (protein kinase B) which are related with NO production. In addition, *Securiniga suffruticosa* increased the protein expression of GTPCH (Guanosine triphosphate cyclohydrolase Ⅰ) and the production of BH4 in HUVEC which are related with eNOS coupling pathway. In conclusion, *Securiniga suffruticosa* has a protective effect against vascular inflammation and can be a potential therapeutic agent for early atherosclerosis.

## 1. Introduction

Vascular inflammation is a condition which plays a key process in the disorder of the endothelium. Endothelium is activated in the inflammatory site where a number of leukocytes adhere to the vascular endothelial layer and infiltrate into the vessel wall and differentiate into macrophages. This is very important in the early steps of development of endothelial dysfunction and tissue damage [1]. Therefore, vascular inflammation and macrophage differentiation are regarded as very important step of endothelial dysfunction [2]. In addition, macrophages can be detected in atherosclerotic lesions, therefore, the production of cytokine can be a reason for atherosclerosis [3,4].

Pro-inflammatory cytokines induce macrophages activation, the expression of adhesion molecules and consequently, the recruitment of inflammatory cells is accelerated [5]. The process of inflammatory cytokine is produced by various cells such as macrophages, lymphocytes, neutrophils and mast cells. Tumor necrosis factor alpha (TNF-α) is very important in this process. TNF-α also can activate both anti- and pro-apoptotic pathways in the myocardium, resulting in an increase of apoptosis, primarily in non-myocytes [6]. TNF-α is resident in atherosclerotic plaque site and increase the expressions of cell adhesion molecules [7]. In addition, cardiovascular diseases commonly reduce oxidative stress which is induced by pathophysiological stimulation such as oxidized low-density lipoprotein [8]. Vascular endothelium can regulate vascular tone, permeability of inflammatory inducers and infiltration of leukocytes [9] and are key factors of the vascular inflammatory process [10]. Cell adhesion molecules (CAMs) also mediate the adhesion and infiltration of macrophages simultaneously [11].

Nuclear factor-kappa B (NF-κB) is a transcription factor in the inflammatory response process. NF-κB is very important for the development of the inflammatory response by up-regulating the expression of many inflammatory mediators that are down-regulated by inhibitory kappa B (IκB) [12]. Transcriptional activation of NF-κB also is a very important process which is related with expression of proinflammatory cell adhesion molecules. Pro inflammatory cytokines mediate the level of reactive oxygen species (ROS) and NF-κB activation can increase the intracellular level of reactive oxygen species (ROS). Therefore, NF-ĸB activation can be inhibited by antioxidants and free radical quenchers [13]. Thus, the main medium of ROS/NF-κB pathway can be a useful treatment approach for vascular diseases including atherosclerosis [14].

Nitric oxide (NO) plays a key role in vascular protection and therapeutic action. NO is widely produced by NO synthase (NOS) and NO is known as the strongest vasodilators. In addition, it is generally known, the levels of NO control the inflammatory symptoms and act in the early stages of inflammation [15,16,17]. Phosphatidylinositol 3′-kinase (PI3K) plays an important role in avoiding cell death via activation of downstream serine/threonine kinase Akt [18]. Akt activates endothelial nitric oxide synthase (eNOS), which produces nitric oxide (NO) [19]. PI3K/Akt/eNOS pathway plays an important role in NO production and preventing endothelial cell injury [20]. In addition, vascular oxidative stress, eNOS uncoupling, damaged aorta NO formation and endothelial dysfunction can be observed in spite of GTPCH protein inducing [21] and tetrahydrobiopterin (BH4) also is a critical cofactor that couples nitric oxide synthase and facilitates the production of nitric oxide (vs. superoxide anions) [22].

*Securinega suffruticosa* is a dioecious shrub belonging to the *Euphorbiaceae* subfamily *Phyllanthoideae*, tribe *Phyllantheae*, subtribe *Phyllanthineae*. *Securinega suffruticosa* grows wild in the south-east Asia, from Korea on the south, to the Caucasus in the northern west, especially in Mandschuria and Internal Mongolia, gardened in Japan, China and Russia (mostly in the European part), Ukraine, rarely in Western Europe and Northern America [23]. *Securinega suffruticosa* was used for the treatment of rheumatic disease, quadriplegia, paralysis following infectious disease, impotence, psychical disorders [24] and children’s malnutrition. In traditional Korean medicine, *Securiniga suffruticosa* has been used for improving the blood circulation and relaxing muscles and tendons, thereby protect and strengthen kidney and spleen. Previous study reported about the antioxidative effect of whole ethanol extract of *Securiniga suffruticosa* [25]. However, there are no reports regarding its vascular protective effect on endothelial cells. Therefore, in this study, treatment of *Securiniga suffruticosa* showed protective effect of inhibiting the vascular inflammation in human umbilical vein endothelial cells (HUVECs) by inducing NO production and eNOS coupling pathway.

## 2. Materials and Methods 

### 2.1. Reagents

TNF-α, antibiotic-antimycotic, fetal bovine serum (FBS), chloromethyl derivative of 2′,7′-dichlorodihydrofluorescein di-acetate (CM-H2DCFDA) and trypsin-ethylenediaminetetraacetic acid (EDTA) were purchased from Invitrogen (Carlsbad, CA, USA). IL-6, VCAM-1, E-selectin, ICAM-1, IκB-α, p-IκB-α, NF-κB, laminB, eNOS, Akt, phospho-eNOS, phosphor-Akt, GTPCH, NLRP3, ASC, caspase-1 and β-actin antibodies were purchased from Santa Cruz Biotechnology (Dallas, TX, USA). Horseradish peroxidase (HRP)-conjugated secondary antibodies were obtained from stress gen Biotechnologies Corp & Enzo Life Sciences (Farmingdale, NY, USA). 2’,7’-bis-(2-carboxyethyl)-5-(and-6)-carboxyfluorescein acetoxymethyl ester (BCECF-AM) was purchased from sigma Chemical co. (St. Louis, MO, USA). Human tetrahydrobiopterin ELISA kit was purchased from MyBioSource (San Diego, CA, USA). The other reagents used in this study were the highest purity commercially available.

### 2.2. Preparation of Securinega Suffruticosa

The dried leaf of *Securinega suffruticosa* was purchased from the Herbal Medicine Co-operative Association, Samcheok, Gangwon Province, Korea. Leaves and twigs of *Securinega suffruticosa* (400 g) was soaked in 4 L and 2 L of 80% ethanol and then sealed with aluminum foil during 1 week at room temperature. After that, boiled at 80–90 °C for 2 h. The extract was filtered three times using gauze and centrifuged at 1910× *g* for 20 min at 4 °C. The filtrate was concentrated using a rotary evaporator (EYELA Rotary vacuum evaporator, Tokyo RIKAKIKAI Co. Ltd., Tokyo, Japan) and lyophilized the resulting extracts (Leaves: 47.014 g, Twigs: 10.766 g) into a freeze-drier (IlshinBioBase, Gyeonggi, Korea). The extract is kept at 4 °C and was dissolved in distilled water before use. In this study, the efficacy was confirmed using the extract of *Securinega suffruticosa* leaves.

### 2.3. HPLC Analysis of Securinega Suffruticosa

High purity reference standards were purchased, namely: (+)-gallocatechin 99.5% from Shanghai Sunny Biotech Co., Ltd. (Shanghai, China), bergenin 99.4% and rutin 95.0% from Merck KGaA (Darmstadt, Germany); (+)-catechin 99.3% and isoquercitrin 99.2% from Biopurify Phytochemicals (Chengdu, China); viroallosecurine 99.0% from ChemNorm Biotech Co., Ltd. (Wuhan, China); and securinine 98.0% from ChemFaces Biochemical Co., Ltd. (Wuhan, China); and securinine (98.0%) was purchased from ChemFaces Biochemical Co., Ltd. (Wuhan, China), respectively. Chemical structures of these reference standard components are shown in Figure 1. High-performance liquid chromatography (HPLC)-grade solvents (methanol, acetonitrile and water) and ACS (American Chemical Society) reagent, ammonium acetate (99.2%) for chromatographic separation and analysis were purchased from J. T. Baker (Phillipsburg, NJ, USA) and Merck KGaA (Darmstadt, Germany), respectively. HPLC analysis for the comparison of the 7 marker components in leaves of *Securinega suffruticosa* was performed using a Shimadzu Prominence LC-20A Series (Kyoto, Japan) consisting of a solvent delivery unit (LC-20AT), online degasser (DGU-20A3), column oven (CTO-20A), auto sample injector (SIL-20AC), photodiode array (PDA) detector (SPD-M20A) and LCSolution software (Version 1.24, SP1, Kyoto, Japan). The 7 reference standards were separated using a Waters SunFireTM C18 analytical column (4.6 × 250 mm, 5 μm; Torrance, CA, USA) maintained at 30 °C and two mobile phases consisting of 20 mM ammonium acetate(A) and acetonitrile (B). The gradient flow of these mobile phases is as follows: 0–20 min, 2–5% B; 20–25 min, 50% B; 25–30 min, 50–2% B; 30–40 min, 2% B. The flow rate of the mobile phase was 1.0 mL/min and injected volume of standard and test solution was 10 μL, respectively. For the quantitative analysis of the 7 reference standards in leaves and twigs of *Securinega suffruticosa*, each test solution of 10.0 mg of leaves and twigs extracts of *Securinega suffruticosa* was dissolved in 10 mL of 70% methanol and ultrasonically extracted for 60 min, respectively. All assay samples were filtered through a 0.2 μm membrane filter (Pall Life Sciences, Ann Arbor, MI, USA) and then injected into the HPLC instrument. Each standard solution of the 7 marker components was prepared at 1000 μg/mL using methanol and then refrigerated.

### 2.4. Cell Cultures

Human umbilical vein endothelial cells (HUVEC) were obtained from Promo Cell (Heidelberg, Germany). Human monocytic leukemia cell line (HL-60) was purchased from ATCC (Manassas, VA, USA). HUVECs and HL-60 were cultured at a density of 5 × 10^5^ cells/mL in Endothelial Cell Basal Medium2 (Promo cell, Heidelberg, Germany) and RPMI 1640 medium (Thermo scientific, Rockford, IL, USA). 10% heat-inactivated FBS and 100 U/mL penicillin G are supplemented in these medium. Cells were incubated at 37 °C in a humidified atmosphere containing 5% CO_2_ and 95% air.

### 2.5. Cell Viability Assay Using MTT

Cytotoxicity was determined by 3-(4,5-dimethylthinazol-2-yl)-2,3-diphenyl tetrazolium bromide (MTT) assay. The HUVEC were seeded onto 96 well culture plates at a density of 1 × 10^4^ cells/well. After incubation with various concentrations of *Securinega suffruticosa* in serum-free Endothelial Cell Basal Medium2 for 24 h, 20 μL of MTT solution (0.5 mg/mL) was added to each culture well and incubated at 37 °C for 4 h more. After incubation, 200 μL of dimethyl sulfoxide (DMSO, Amresco Inc., Dallas, TX, USA) was added and dissolved the crystals for 10 min. MTT reduction was quantified by measuring the light absorbance with a multilabel plate reader (F-2500, Hitachi, Tokyo, Japan) and the absorbance was used as a measurement of cell viability which control group was considered 100% viable.

### 2.6. Monocyte-HUVEC Adhesion Assay

HUVECs were grown to confluence in 6-well culture plates, pretreated with *Securinega suffruticosa* for 30 min and stimulated with Ox-LDL for 6 h. HL-60 cells were labeled with 10 μM BCECF-AM for 1 h at 37 °C and washed twice with growth medium. HL-60 cells bound to HUVECs were detected by fluorescence microscopy and then lysed with 50 mM Tris-HCl (Trishydrochloride), pH 8.0, containing 0.1% SDS (Sodium dodecyl sulfate). Monocyte adhesion was visualized using fluorescence microscopy (Nikon Eclipse Ti, Tokyo, Japan). Fluorescence intensity was measured using a spectrofluorometer (F-2500, Hitachi, Tokyo, Japan) at excitation and emission wavelengths of 485 and 535 nm, respectively.

### 2.7. Western Blot Analysis

Cell homogenates were loaded onto 8–12% running gels, separated via SDS-polyacrylamide gel electrophoresis (SDS-PAGE) and then transferred to a nitrocellulose membrane. Membranes were then washed with H_2_O and blocked with 5% FBS in tris-buffered saline with 0.05% Tween-20 (TBS-T) for 2 h and incubated with primary antibodies (overnight, 4 °C). The membranes were then washed with TBS-T and incubated with secondary antibodies conjugated to horseradish peroxidase (HRP) for 1 h at room temperature (25 °C). Protein bands were visualized using iBright FL-1000 (Invitrogen, Carlsbad, CA, USA). Densitometry analysis of protein bands was conducted with the ImageJ (NIH, Bethesda, MD, USA) program.

### 2.8. RNA Preparation and Quantitative Real-Time Reverse Transcription-PCR (Real-Time RT-qPCR)

Total cellular RNA was extracted from HUVECs using the trizol reagent. The cDNA was synthesized at 500 ng mRNA through 20 μL reverse transcription reaction cultured in the SimpliAmp Thermal Cycler (Life technology, Carlsbad, CA, USA). The samples were incubated at 42 °C for 60 min, 94 °C for 5 min and then the cDNA was used and immediately stored at −20 °C. The sequences of primers and probes were as follows: VCAM-1 (forward: 5′-ATG CCT GGG AAG ATG GTC GTG A-3′, reverse: 5′-TGG AGC TGG TAG ACC CTC GCT G-3′), ICAM-1 (forward: 5′-AGG CCA CCC CAG AGG ACA AC-3′, reverse: 5′-CCC ATT ATG ACT GCG GCT GCT A-3′) and IL-6 (forward: 5′-GAA CTC CTT CTC CAC AAG CGC CTT-3′, reverse: 5′-CAA AAG ACC AGT GAT GAT TTT CAC CAG G-3′). The real-time RT-qPCR (Quntitative reverse transcription polymerase chain reaction) carried in a 20-μL final volume containing 1 μL cDNA sample, 1 μL primer pairs, 8 μL ultra-pure distilled water and 10 μL SYBR Green PCR Master Mix and performed by an initial denaturation step at 95 °C for 10 min, followed by 40 cycles at 95 °C for 15 s and finally 60 °C for 60 s in the Step-One™ Real-Time PCR System (Applied Biosystems, Carlsbad, CA, USA). Each RNA sample was measured in triplicate. The resulting of mRNA abundance data was normalized against GAPDH mRNA abundance.

### 2.9. Preparation of Cytoplasmic and Nuclear Extracts

The cells were rapidly harvested and nuclear and cytoplasmic extracts can be obtained by using the nuclear and cytoplasmic extraction reagents kit^®^ (Thermo scientific, Rockford, IL, USA) according to the manufacturer’s protocol. Briefly, cells were harvested and resuspended in ice-cold cytoplasmic extraction reagent-I and add cytoplasmic extraction reagent-II^®^ to mixture and centrifuged at 16,700× *g* with 20 min. Upper solution was moved to asepsis tube and used as cytoplasmic extract. After then, subsequently nuclear extraction reagent was added remaining pellet and blended at intervals of 30 min and then centrifuged at 16,700× *g* for 10 min. The supernatant was used as nucleus extract.

### 2.10. Immunofluorescence Microscopy 

For localization of NF-κB p65, HUVECs were grown on Lab-Tek II chamber slides, treated with *Securinega suffruticosa* for 30 min and incubated with TNF-α (50 ng/mL) for an additional hour. Cells were fixed with 4% paraformaldehyde and were treated with 0.1% Triton X-100 for permeability. The cells were detected using NF-κB p65 antibody and fluorescein isothiocyanate (FITC)-labeled secondary antibody (Santa Cruz, CA, USA). To visualize the nuclei, cells were then treated with 1 µg/mL of DAPI (4′,6-diamidino-2-phenylindole) for 30 min. Cells were washed with PBS and coverslips were mounted with mounting solution onto glass slides; the slides were then examined under a fluorescent microscope (Nikon Eclipse Ti, Tokyo, Japan).

### 2.11. Intracellular ROS Production Assay

CM-H2DCFDA was used to detect the intracellular ROS as fluorescent probe. Briefly, the confluent HUVECs in the 60 mm^2^ dish culture plates were pretreated with *Securinega suffruticosa* for 30 min. After removal from the wells, they were incubated with 1 μM CM-H2DCFDA for 1 h and then stimulated using TNF-α. Intracellular ROS was visualized using fluorescence microscopy (Nikon Eclipse Ti, Tokyo, Japan). The fluorescence intensity was measured using a spectrofluorometer (F-2500, Hitachi, Tokyo, Japan) at excitation and emission wavelengths of 485 and 535 nm, respectively.

### 2.12. Measurement of Nitrite Production Using Griess Reagent System

NO production was evaluated spectrophotometrically by measuring nitrite which is an oxidative product of NO. Nitrite levels were examined by using the Griess Reagent System (Promega, Madison, WI, USA). The fluorescent intensity was then measured with an Infinite F200 pro fluorometer (Tecan, Männedorf, Switzerland) at 485 and 535 nm.

### 2.13. Fluorescence Microscopy of Intracellular NO Generation

Intracellular NO generation was detected with NO-sensitive fluorescence dye DAF-2DA (Merck Biosciences, Schwalbach, Germany). Briefly, HUVECs were seeded in a 6 well plate, grown to confluence and incubated with *Securinega suffruticosa* for 30 min. DAF-2DA was added 30 min before stopping the reactions and immediately after the cells were fixed in 2% of paraformaldehyde for 30 min at room temperature. The intensity of fluorescence was examined under a fluorescent microscope (Nikon Eclipse Ti, Tokyo, Nikon).

### 2.14. Measurement of BH4 Using ELISA Kit

Human tetrahydrobiopterin level in the HUVEC was spectrophotometrically evaluated by measuring BH4 expression in cell homogenates. BH4 levels was determined with the Human Tetrahydrobiopterin ELISA (Enzyme-linked immunosorbent assay) kit (MyBioSource, San Diego, CA, USA). The absorbance was read at 450 nm using a microplate reader (Infinite F200 pro, Tecan, Switzerland).

### 2.15. Statistical Analysis

All experiments were repeated at least three times. Results were expressed as mean ± standard error (S.E.) and data were analyzed using one-way analysis of variance followed by a Student’s *t*-test via sigmaplot software (version 10.0, SYSTAT, San Jose, CA, USA) to determine any significant differences. Differences were considered statistically significant when *p* values were less than 0.05, 0.01 and 0.001.

## 3. Results

### 3.1. HPLC Analysis of Securinega Suffruticosa

The established HPLC assay was used for the simultaneous analysis. There were four flavonoids ((+)-gallocatechin, (+)-catechin, rutin and isoquercitrin), one phenol (bergenin) and two alkaloids (viroallosecurinine and securinine) in *Securinega suffruticosa*. All marker components were separated with a resolution >4.0 within 25 min (Figure 2). Each component was detected at 255 nm (viroallosecurinine and securinine), 270 nm ((+)-gallocatechin and bergeninA), 280 nm ((+)-catechin) and 355 nm (rutin and isoquercitrin) based on the λmax of UV spectrum for the simultaneous quantification. All validation data of HPLC assay, namely, linear range, regression equation, coefficient of determination (*r*^2^), limit of detection (LOD) and limit of quantification (LOQ) are presented in Table 1. Using established HPLC assay, the 7 marker components in the leaves of *Securinega suffruticosa* were detected as 1.01–20.51 mg/g. On the other hand, rutin, isoquercitrin and securinine were not detected in the twigs of *Securinega suffruticosa* but the other four components ((+)-gallocatechin, bergenin, (+)-catechin and viroallosecurinine) were detected 4.42–141.34 mg/g. Detailed content results for each marker component in leaves and twigs of *Securinega suffruticosa* are shown in Table 2. In previous study, securinine has a protective effect in vascular endothelial cells [26]. However, there was no securinine in twigs of *Securinega suffruticosa.* Therefore, the leaves of *Securinega suffruticosa* was used for this study.

### 3.2. Cytotoxicity of Securinega Suffruticosa in HUVEC

Cytotoxicity effect of *Securinega suffruticosa* on HUVEC was evaluated by MTT cytotoxicity assay. Figure 3 shows that *Securinega suffruticosa* did not affect the cell viability up to 100 µg/mL (>100% cell viability). Therefore, 50 µg/mL of *Securinega suffruticosa* was used as the maximum concentration in this study.

### 3.3. Effect of Securinega Suffruticosa on TNF-α-Induced VCAM-1, ICAM-1 and IL-6 Expressions in HUVEC

The effect of *Securinega suffruticosa* on the expression of the cell adhesion molecules in HUVEC was confirmed by performing Western blot analysis. *Securinega suffruticosa* significantly inhibited the TNF-α-induced VCAM-1, ICAM-1 and IL-6 expressions (*p* < 0.05) (Figure 4). In addition, real-time RT-qPCR analysis was performed in the presence of TNF-α or without *Securinega suffruticosa* to confirm whether *Securinega suffruticosa* inhibits VCAM-1 production at the mRNA expression levels. As shown in Figure 5, TNF-α-induced obvious increasing of VCAM-1, ICAM-1 and IL-6 mRNA expressions, whereas pretreatment with *Securinega suffruticosa* (10–50 µg/mL) was markedly decreased VCAM-1 mRNA expressions in a dose-dependent manner respectively.

### 3.4. Effect of Securinega Suffruticosa on TNF-α-Induced NF-κB p65 Expression 

Cytokines activate NF-κB phosphorylation and IκB-α degradation. Translocation of p65 subunit from cytoplasm to the nucleus is needed for NF-κB activation, where it acts as a transcription factor for genes related to cell growth. Therefore, in this study, the effect of *Securinega suffruticosa* on activation and translocation of NF-κB were examined. Pretreatment of cells with *Securinega suffruticosa* inhibited phosphorylation of IκB-α in cytoplasmic extract (CE). *Securinega suffruticosa* also inhibited NF-κB p65 levels in nucleus extract (NE) (Figure 6A). Immunofluorescence was performed using NF-κB p65 and FITC (Fluorescein)-conjugated antibody to confirm the result of the Western blot analysis. As a result, TNF-α increased NF-κB p65 translocation to the nucleus, on the contrary this translocation was inhibited by treatment with *Securinega suffruticosa* in a dose-dependent manner (Figure 6B).

### 3.5. Inhibitory Effect of Securinega Suffruticosa on TNF-α-Induced ROS Production

To clarify the inhibitory effect of *Securinega suffruticosa* on TNF-α-induced oxidative stress, HUVECs were labeled with CM-H2DCFDA. And the level of ROS production was visualized with fluorescence microscope (Figure 7A). The intensity of fluorescence was analyzed with a spectrofluorometer (Figure 7B). In Figure 7, TNF-α induced intracellular ROS levels compared to the untreated cells. However, pretreatment with *Securinega suffruticosa* significantly reduced the TNF-α-induced ROS levels. In addition, NAC (10 mM) was used as a positive control to confirm the inhibitory effect of *Securinega suffruticosa* more precisely.

### 3.6. Effect of Securinega Suffruticosa on Ox-LDL-Induced Adhesion of HL-60 Cell to HUVEC

Cell adhesions between HL-60 cells and Ox-LDL-stimulated HUVEC were detected to confirm the effect of *Securinega suffruticosa*. HUVEC was pretreated with *Securinega suffruticosa* (10–50 µg/mL) and then stimulated with Ox-LDL for 6 h. As shown in Figure 8, when HUVEC was not treated with Ox-LDL, minimal binding to HL-60 cells was detected. However, adhesion was significantly increased by Ox-LDL stimulation. Pretreatment with (10–50 μg/mL) was decreased the number of HL-60 cells adhering to Ox-LDL-induced HUVEC. Specially, when treated with 50 µg/mL, the number of HL-60 cells adhesion significantly decreased. Therefore, *Securinega suffruticosa* has the effective to prevent the early process of vascular inflammation.

### 3.7. Securinega Suffruticosa Stimulates the Production of NO in HUVEC

NO is a relaxation factor derived from the endothelium and plays a key role in vessel tone and function regulation. The phosphorylation of eNOS and Akt simulates NO production in endothelial cells. In this study, the level of NO produced by HUVECs was measured when treated with different concentrations of *Securinega suffruticosa*. Figure 9 shows that *Securinega suffruticosa* increased the production of NO in HUVECs dose-dependently.

### 3.8. Securinega Suffruticosa Stimulates the eNOS Coupling in HUVEC

This study was investigated to confirm the effect of *Securinega suffruticosa* on the expressions of some phosphorylated proteins which is linked with eNOS coupling such as p-endothelial cell nitric oxide synthase (p-eNOS, p-NOS3) and p-Akt in endothelial cells. For the dose-dependent experiment, endothelial cells were treated with different doses of *Securinega suffruticosa* for 30 min and harvested for western blot analysis. In addition, L-NAME (N(ω)-nitro-L-arginine methyl ester) and ADMA as inhibitor of NOS3 and LY294002 and wortmannin as inhibitor of Akt each were treated. As shown in Figure 10A, the expressions of phosphorylated eNOS and Akt increased in dose-dependent manner, compared to the total forms of the respective proteins. These results suggest that *Securinega suffruticosa* stimulates p-eNOS and p-Akt expression in HUVECs. Figure 10B shows that *Securinega suffruticosa* treatment dose-dependently increased the expression of GTPCH and the production of BH4 in HUVECs. In order to investigate the relationship between NO production and inflammation, cell adhesion molecule protein expressions were measured after treated with inhibitors of eNOS pathway such as L-NAME (N(ω)-nitro-L-arginine methyl ester) and. As shown in Figure 10C, the decreasing of expressions of cell adhesion molecules were inhibited by each inhibitor.

## 4. Discussion

The leaf and twig of *Securiniga suffruticosa* has been used as medicinal plant for improving the blood circulation and relaxing muscles. This study provides the first evidence suggesting that *Securiniga suffruticosa* has a protective effect by inhibiting TNF-α-induced adhesion molecules and inducing NO production in HUVEC.

Vascular inflammation is related with pathogenesis of atherosclerosis. Many previous studies reported about importance of adhesion molecules in atherosclerosis. The interaction between monocytes and molecules which is expressed on the endothelial cell surface mediate this inflammatory step [27]. At first, this study found whether *Securiniga suffruticosa* suppressed monocytes adhesion to TNF-α-stimulated HUVEC by suppressing expression of cell adhesion molecules and pro-inflammatory cytokines such as VCAM-1, ICAM-1 and IL-6. The expression of these CAMs on the endothelial cell membrane are increased by stimulation of cytokines such as TNF-α [28]. Cytokines and adhesion molecules are well known for inflammation markers that are closely related with the development and pathogenesis through microvascular and inflammatory mechanisms [29]. Therefore, in this study, the expression of cell adhesion molecules and cytokine were examined to confirm whether *Securiniga suffruticosa* has an inhibitory effect on the adhesion monocytes to TNF-α-stimulated HUVECs by inhibiting the protein expressions of adhesion molecules and cytokines such as VCAM-1, ICAM-1 and IL-6. In addition, these can mediate the adhesion of monocytes to the endothelium in atherosclerotic lesions. Above all, this study showed that *Securiniga suffruticosa* attenuated expression level of VCAM-1, ICAM-1 and IL-6. These results suggest that *Securiniga suffruticosa* has protective effect on TNF-α-induced vascular inflammatory process in endothelial cells.

NF-κB exists in the cytoplasm in unstimulated condition as a heterodimer composed of two subunits, p50 and p65, bound to the inhibitory protein IκB-α. However, when cells are stimulated, IκB-α is phosphorylated and degraded, allowing NF-κB to be translocated into nucleus and start gene transcription [30]. To investigate the anti-inflammatory effect of *Securiniga suffruticosa* on TNF-α-stimulated HUVECs, the protein expression of NF-κB in nucleus and IκB-α in the cytosol was examined by using western blot analysis and translocation of NF-κB into nucleus by using immunofluorescence microscopy. As a result, *Securiniga suffruticosa* inhibited TNF-α-induced degradation of IκB-α, phosphorylation of NF-κB p65 and nuclear translocation of NF-κB p65.

Oxidative stress occurs in the early stages of atherosclerosis and plays a key role in the development of vascular inflammation [31]. The ROS/NF-κB pathway is a key mediator which regulates inflammatory responses to vascular diseases. ROS are essential to the functions of cells but overproduction can have harmful effects through NF-κB pathway [32]. Therefore, the production of ROS in HUVECs was quantified by measuring the amount of CM-H2DCFDA and visualized via immunofluorescence microscopy. TNF-α-induced elevations in intracellular ROS levels were significantly reversed by pretreatment with *Securiniga suffruticosa*.

Endothelial dysfunction, characterized by reducing NO production, is an early key medium linking in cardiovascular diseases [33]. Nitric oxide, a vasodilator synthesized by eNOS, protects blood vessels by inhibiting platelet aggregation, monocyte adhesion and smooth muscle cell proliferation [34]. Akt, downstream of PI3K, is also considered as a key factor in cell survival. In endothelial cells, cell survival is promoted by Akt activation [19]. Tetrahydrobiopterin (BH4) is natural co-factor of the three aromatic amino acid hydroxylases. In addition, BH4 is very important regulatory factor in the biosynthesis of neurotransmitters [35]. BH4 is a co-factor of isoforms of nitric oxide synthase (NOS) and is necessary for catalyst activation [36]. eNOS uncoupling into superoxide production is caused by BH4 deficiency [37]. GTP cyclohydrolase 1 (GTPCH1) is a kind of enzyme in the de novo biosynthetic pathway of BH4. GTPCH1 suppression causes the BH4 level to decrease sharply and consequently eNOS to be uncoupled [38]. GTPCH1 deficiency can be an important cause of endothelial dysfunction in cardiovascular diseases and diabetes [39]. Data of this study show that, *Securiniga suffruticosa* enhanced NO production. In addition, this study also indicated that *Securiniga suffruticosa* has an anti-inflammatory effect by regulating the activation of PI3K/Akt pathway and phosphorylation of eNOS. eNOS and Akt phosphorylation were also increased by *Securiniga suffruticosa*. *Securiniga suffruticosa* also increased the levels of GTPCH and BH4. In addition, the inhibition of the expression of cell adhesion molecules by *Securiniga suffruticosa* was suppressed by inhibitors of eNOS pathway. Taken together, these results provide strong evidence that *Securiniga suffruticosa* has an anti-inflammatory effect through PI3K/Akt-dependent eNOS and eNOS coupling pathways.

In conclusion, *Securiniga suffruticosa* suppresses vascular inflammatory process and enhancing PI3K/Akt/eNOS pathway. However, further study will be needed to detect the expression of relevant factors to determine the efficacy in later stages of atherosclerosis. In addition, in vivo study also needs to be carried out to prove the effectiveness of *Securiniga suffruticosa* more clearly.

## 5. Conclusions

*Securiniga suffruticosa* treatment has significantly lowered the process of vascular dysfunction and inflammation. *Securiniga suffruticosa* treatment ameliorated impairment of vascular dysfunction and up-regulated eNOS coupling pathway in HUVECs (Figure 11). In conclusion, these findings suggest the first evidence supporting the therapeutic efficacy of *Securiniga suffruticosa* in preventing the development of vascular inflammation and atherosclerosis.

## Figures and Tables

**Figure 1 nutrients-12-03448-f001:**
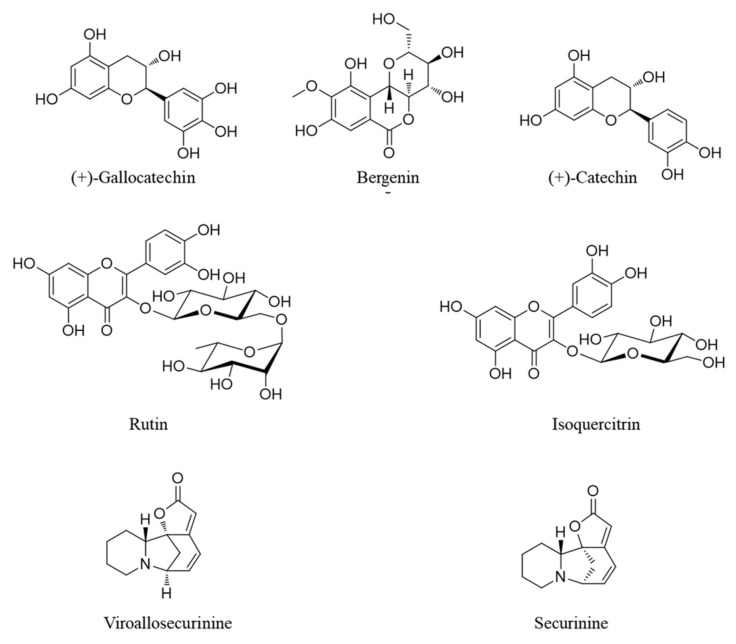
Chemical structures of seven marker components in leaves and twigs of Securinega suffruticosa.

**Figure 2 nutrients-12-03448-f002:**
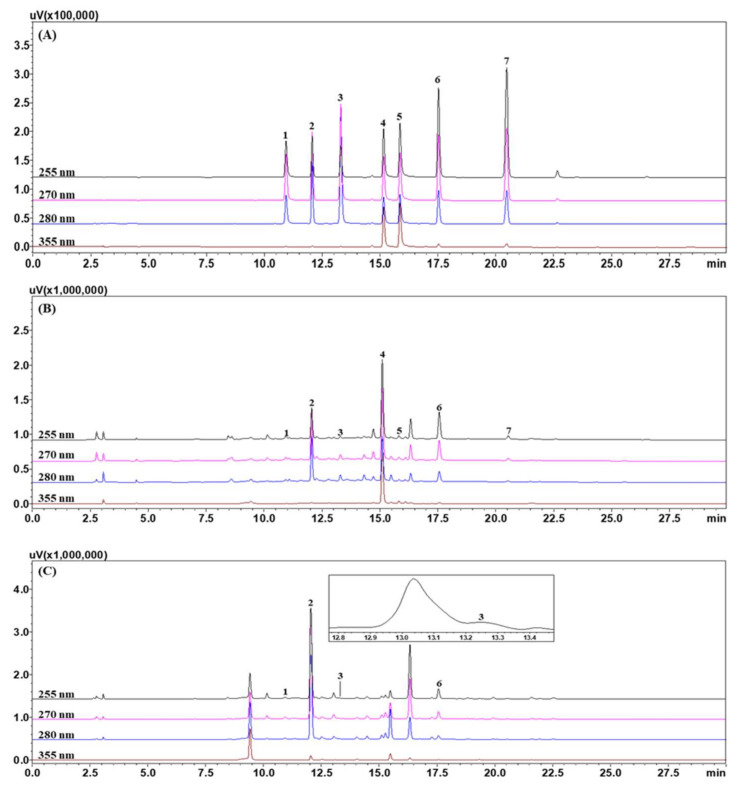
HPLC (High performance liquid chromatography) chromatograms of the standard solution (**A**), leaves (**B**) and twigs (**C**) of *Securinega suffruticosa*. (+)-Gallocatechin (1), bergenin (2), (+)-catechin (3), rutin (4), isoquercitrin (5), viroallosecurinine (6) and securinine (7).

**Figure 3 nutrients-12-03448-f003:**
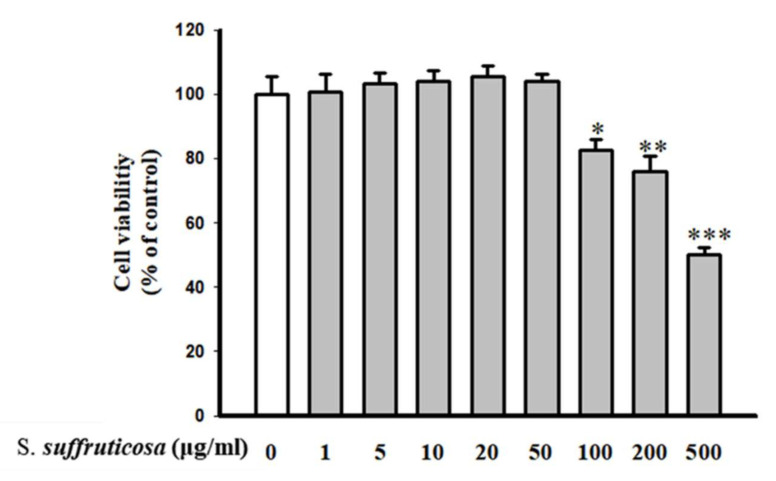
Effect of *Securinega suffruticosa* on cell viability of human umbilical vein endothelial cells (HUVECs). Cells were treated with leaf extract of *Securinega suffruticosa* (0–500 μg/mL) for 24 h. The data are expressed as a percentage of basal value and are the means ± S.E. of five independent experiments with four dishes. * *p* < 0.05, ** *p* < 0.01, *** *p* < 0.001 versus control.

**Figure 4 nutrients-12-03448-f004:**
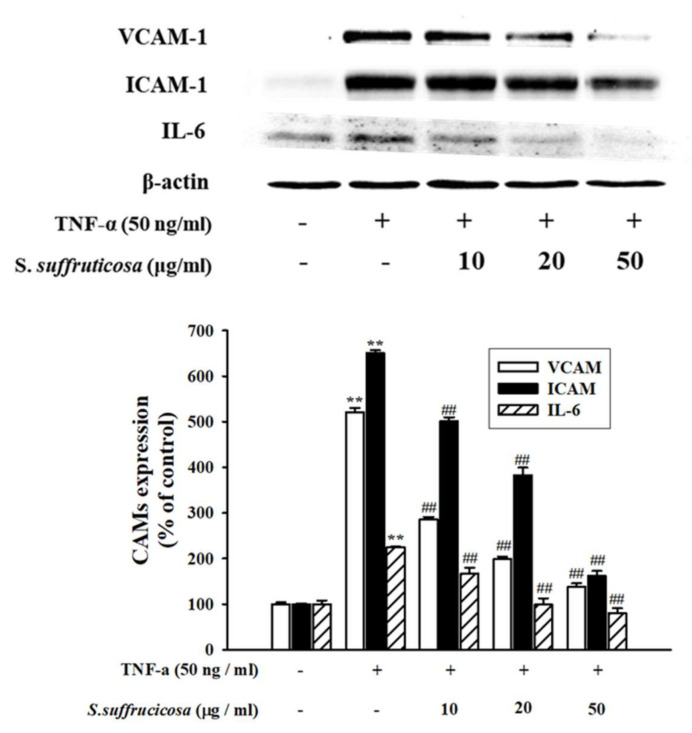
Effect of *Securinega suffruticosa* on TNF-α-induced vascular inflammation related molecule expression. The HUVEC were pretreated with *Securinega suffruticosa* for 30 min and followed by stimulation with TNF-α (50 ng/mL) for 24 h. VCAM-1, ICAM-1 and IL-6 protein expressions were analyzed by Western blotting. ** *p* < 0.01 vs. control, ## *p* < 0.01 versus TNF-α alone.

**Figure 5 nutrients-12-03448-f005:**
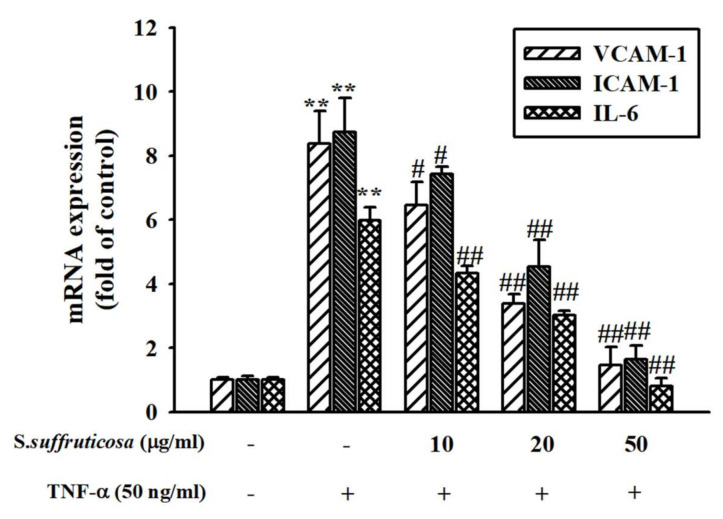
Effect of *Securinega suffruticosa* on TNF-α-induced vascular inflammation related molecule mRNA expression. HUVEC were pretreated with *Securinega suffruticosa* (10–50 µg/mL) for 30 min and then stimulated with TNF-α (50 ng/mL) for 24 h. VCAM-1, ICAM-1 and IL-6 mRNA expressions were analyzed by using real-time RT-qPCR (Quntitative reverse transcription polymerase chain reaction). ** *p* < 0.01 vs. non-treated control group, # *p* < 0.05, ## *p* < 0.01 vs. TNF-α treated group.

**Figure 6 nutrients-12-03448-f006:**
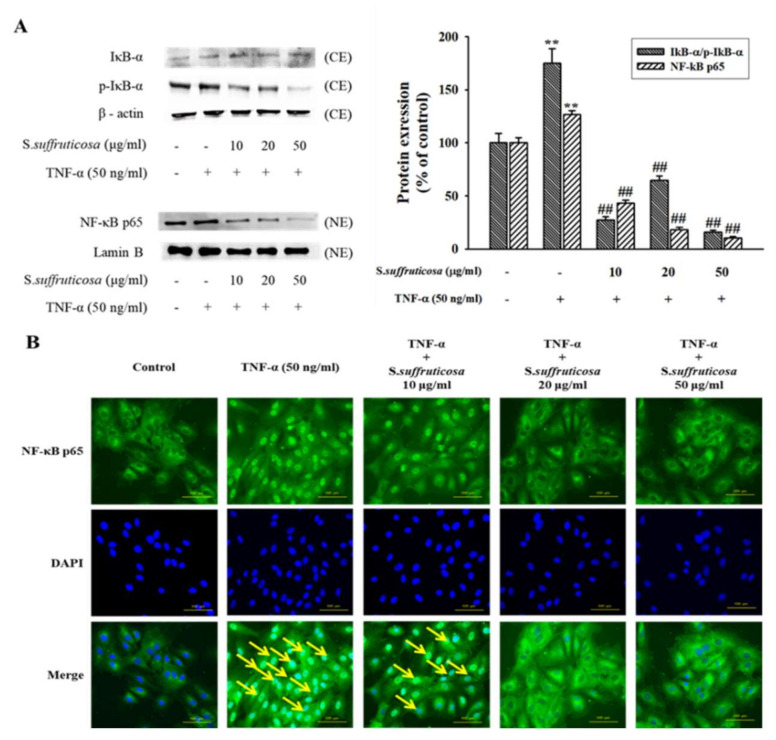
Effects of *Securinega suffruticosa* on TNF-α-induced NF-κB p65 expression. The nuclear protein extract (NE) and cytoplasmic protein extract (CE) were prepared and separated on 10% SDS (Sodium dodecyl sulfate)-PAGE (Polyacrylamide gel electrophoresis) and blotted with the antibodies specific for IκB-α, p-IκB-α and NF-κB p65 (**A**). Immunostaining of NF-κB p65 by using a monoclonal antibody (green), nuclei stained with DAPI (blue) and composite images generated by superimposing photographs. Each photograph is representative of the results from five independent experiments (original magnification ×400) (**B**). ** *p* < 0.01 vs. control, ## *p* < 0.01 versus. TNF-α alone.

**Figure 7 nutrients-12-03448-f007:**
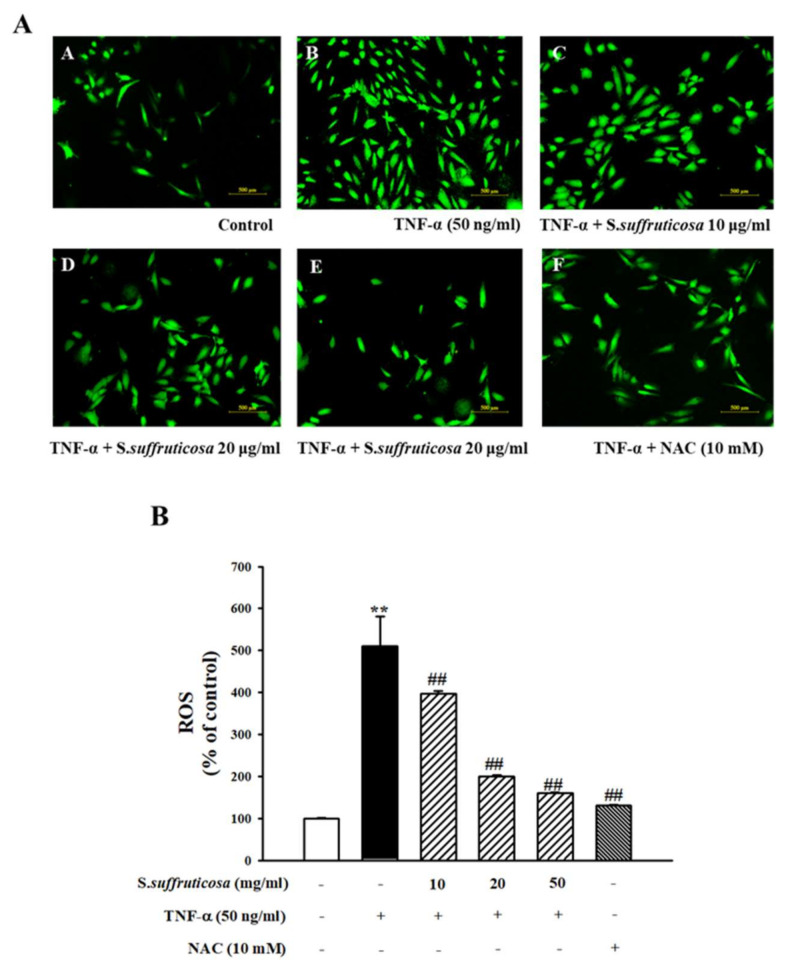
Inhibitory effect of *Securinega suffruticosa* on TNF-α-induced reactive oxygen species (ROS) production. ROS production was examined via fluorescence microscopy (original magnification ×100) (**A**). ROS production was expressed as mean ** *p* < 0.01 versus control, ## *p* < 0.01 versus TNF-α alone (**B**).

**Figure 8 nutrients-12-03448-f008:**
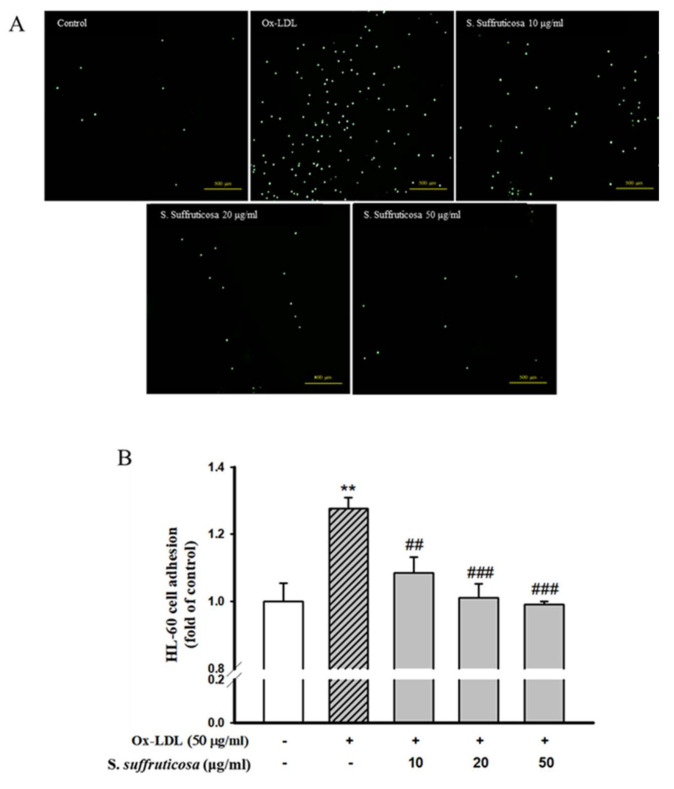
Effect of Securinega suffruticosa on adhesion of monocyte to Ox-LDL-stimulated HUVEC. (**A**) Adherent HL-60 cells visualized using fluorescence microscopy. (**B**) The intensity of fluorescence was measured using a spectrofluorometer. Data are presented as mean ± S.E. of five independent experiments with triplicate dishes. ** *p* < 0.01 vs. control, ## *p* < 0.01, ### *p* < 0.001 vs. Ox-LDL alone.

**Figure 9 nutrients-12-03448-f009:**
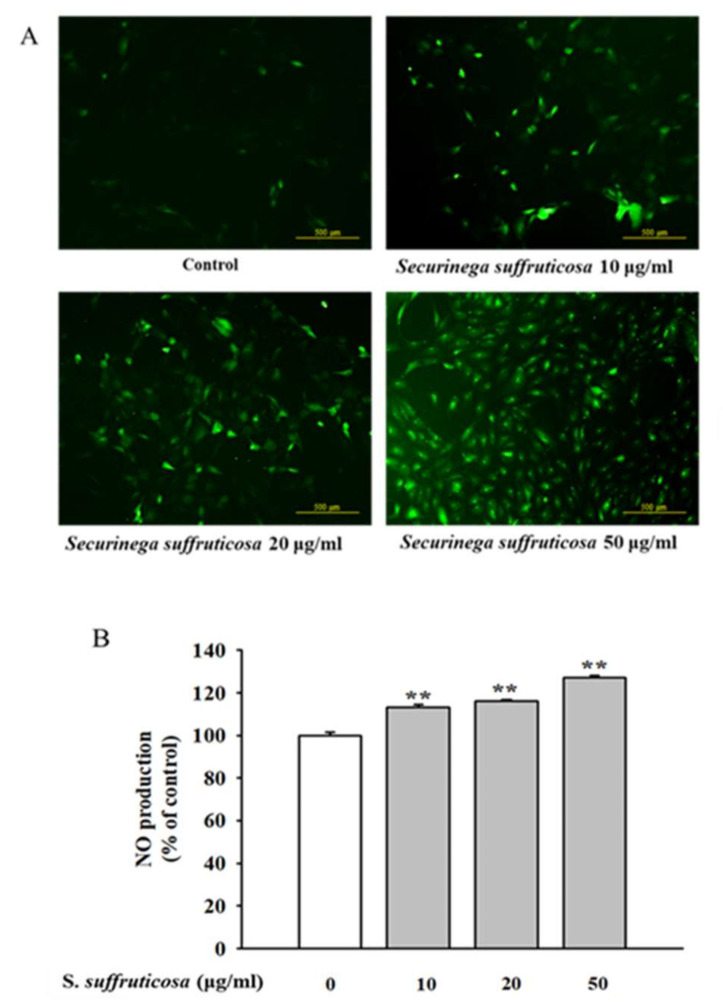
Effect of *Securinega suffruticosa* on nitrite production. (**A**) NO production was examined via fluorescence microscopy (original magnification ×100). (**B**) Effect of *Securinega suffruticosa* on NO production assayed as its stable reaction product nitrite with the Griess reaction. ** *p* < 0.01 vs. control.

**Figure 10 nutrients-12-03448-f010:**
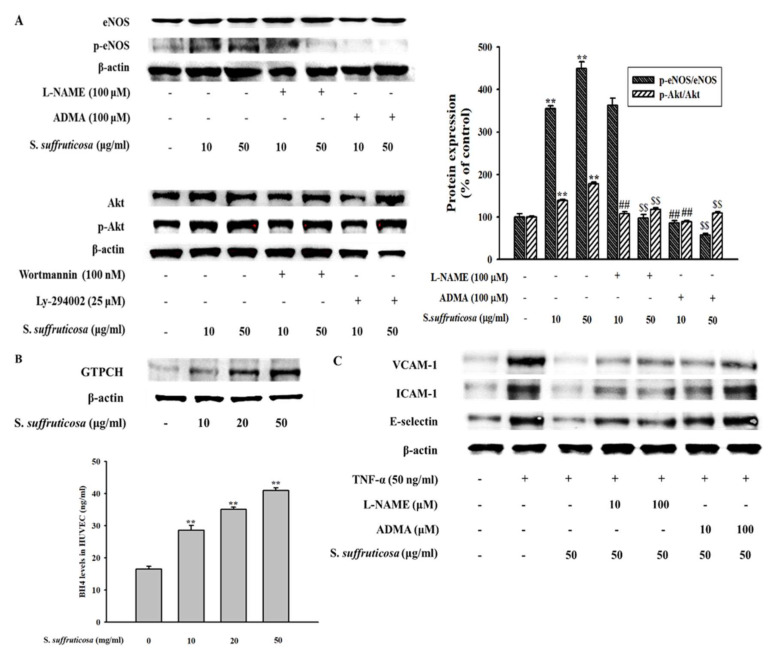
Effect of *Securinega suffruticosa* on eNOS coupling. (**A**) *Securiniga suffruticosa* was treated with various concentrations and protein expression of phosphorylated NOS3 and phosphorylated Akt were analyzed by western blot. (**B**) The total protein extracts of GTPCH and intracellular BH4 level were measured. (**C**) Western blot analysis of VCAM-1, ICMA-1 and E-selectin protein expression. HUVECs were pretreated with different concentrations (10–50 μg/mL) of *Securiniga suffruticosa* for 30 min and treated L-NAME and ADMA during additional 30 min, after then stimulated with TNF-α for 24 h. The whole protein extracts were prepared and separated on 10% SDS-PAGE and blotted with the antibodies specific for VCAM-1, ICMA-1 and E-selectin. ** *p* < 0.01 vs. control, ## *p* < 0.05 vs. 10 μg/mL *Securinega suffruticosa* treated group, $$ *p* < 0.05 vs. 50 μg/mL *Securinega suffruticosa* treated group.

**Figure 11 nutrients-12-03448-f011:**
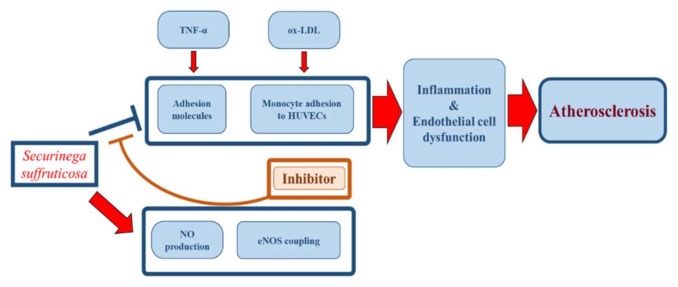
Schematic diagram of the effects of *Securiniga suffruticosa* in vascular inflammation.

**Table 1 nutrients-12-03448-t001:** Linear range, regression equation, *r*^2^, LOD (Limit of detection) and LOQ (Limit of quantitation) for marker compounds (*n* = 3).

Compound	Linear Range (μg/mL)	Regression Equation (*y* = ax + b) ^a^	*r* ^2^	LOD (μg/mL) ^b^	LOQ (μg/mL) ^c^
(+)-Gallocatechin	1.56–100.00	*y* = 2693.75x *−* 2339.14	0.9999	0.30	0.90
Bergenin	3.13–200.00	*y* = 14,964.55x + 8789.50	1.0000	0.19	0.58
(+)-Catechin	1.56–100.00	*y* = 13,587.19x *−* 2117.38	0.9999	0.18	0.55
Rutin	1.56–100.00	*y* = 20,362.88x *−* 7479.16	0.9999	0.11	0.33
Isoquercitrin	0.78–50.00	*y* = 27,537.45x *−* 5765.05	0.9999	0.13	0.41
Viroallosecurinine	0.78–50.00	*y* = 39,678.55x + 3029.15	1.0000	0.15	0.45
Securinine	0.78–50.00	*y* = 41,192.35x + 3453.01	1.0000	0.06	0.19

^a^ y: peak area of compounds; x: concentration (μg/mL) of compounds; ^b^ LOD = 3.3 × σ/S.; ^c^ LOQ = 10 × σ/S (σ: the standard deviation of the y-intercept; S: the slope of the calibration curve).

**Table 2 nutrients-12-03448-t002:** Amount of the 7 marker components in leaves and twigs of *Securinega suffruticosa* (*n* = 3).

Compound	Leaves		Twigs	
	Mean (mg/g) ± SD (× 10^−1^)	RSD (%)	Mean (mg/g) ± SD	RSD (%)
(+)-Gallocatechin	8.51 ± 1.75	2.06	17.84 ± 0.20	1.14
Bergenin	20.51 ± 0.36	0.17	141.34 ± 0.36	0.26
(+)-Catechin	6.75 ± 0.27	0.40	1.61 ± 0.06	3.47
Rutin	20.35 ± 1.66	0.81	ND	-
Isoquercitrin	1.01 ± 0.01	0.11	ND	-
Viroallosecurinine	5.81 ± 1.17	2.01	4.42 ± 0.03	0.68
Securinine	1.46 ± 0.11	0.78	ND	-

RSD: relative standard deviation, ND: not detected.

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
