# Peer review of "Anti-Vascular Inflammatory Effect of Ethanol Extract from Securinega suffruticosa in Human Umbilical Vein Endothelial Cells"

_nutrients, 2020, doi:10.3390/nu12113448_

Round 1
Reviewer 1 Report
In this paper, the authors examine the ability of an extract derived from the plant Securiniga suffruticosa to affect TNFa-induced endothelial cell activation, adhesion molecule expression, monocyte adherence, and related signaling processes. They report that the extract is able to exert an anti-inflammatory effect on endothelial cells via blocking TNFa signaling and increasing NO signaling, and conclude that it has potential as an anti-atherosclerotic agent. Overall, this is an interesting and well conducted study. Comments and suggestions for the authors are listed below.
Introduction/Discussion. Securiniga suffruticosa – some more background information introducing this to readers who are not familiar with it would be helpful. What type of plant is it, where does it grow, what is already known about its medicinal properties, etc. Also, is the material being studied a whole extract of the plant or some more purified preparation of the material (this can be briefly indicated in the abstract/introduction and described in more detail in the Methods/Discussion).
Methods. Line 143 – supplier for endothelial cell culture media?
Results.
Line 235 – seems to be too many parentheses here
It is a little unclear why both leaves and twigs were examined (Table 2 and related text) – a brief rationale for this should be included
3.2 heading should be reworded to “Cytotoxicity of Securiniga suffruticosa in HUVEC” or similar
Figure 5 – why was VCAM mRNA after TNFa +/- Securiniga suffruticosa measured but not ICAM or IL-6?
Figure 6 and elsewhere – abbreviations should be defined in the figure legend (CE, NE)
Figure 6A were the WB data quantified? In the blot presented, it looks like TNFa alone does not really affect either IK-kB or NF-kB, but that Securiniga suffruticosa decreases them both to below baseline levels. However, the text indicates that there is an effect of TNFa, which is blocked by Securiniga suffruticosa. (see also Discussion lines 399-400). The immunofluorescence images in 6B support this conclusion better than the WB which is shown.
Figure 6B, 8A: need a scale bar.
Figure 7A, 8A, 9A: transmitted light images of the same fields would be helpful to confirm that there are equivalent numbers of cells in each field.
Figure 8A please define IYC.
Figure 9 the effect of Securiniga suffruticosa appears larger in the fluorescence images than in the quantification of the reaction assay data below. Can the authors explain why this may be the case?
Figure 10A the blots for p-eNOS and Akt are not very convincing. It looks like p-eNOS is increased in the L-NAME + 10 ug/mL Securiniga suffruticosa group. Akt/-pAkt does not really look different across the treatment groups. Were the WB data quantified?
Figure 10B what is GTPCH? (defined in the Discussion lines 416-417 – please move this to the relevant section of the Results)
Discussion.
Please summarize previously published work on Securiniga suffruticosa related to the present study.
How do the authors envision Securiniga suffruticosa being used clinically, since atherosclerosis is a long-term/chronic disease process? When in this disease process would the treatment potentially be useful?
Please comment on the limitations of the use of HUVEC as a model system (as opposed to primary human aortic endothelial cells, coronary artery endothelial cells, etc.).
Reviewer 2 Report
I am submitting my comments as a pdf file

Round 2
Reviewer 2 Report
The authors have addressed all my comments regarding grammatical and typos mistakes. However, the comments in connection with experimental details were ignored. For example, in my first report I mentioned that there is no information about the reasons behind the selection of the concentration ranges indicated in Table 1, or how the calibration curves (in blank leaves? or in blank twigs?) were prepared, or how the real samples (both leaves and twigs) were treated, or why the calibration curves were prepared in μg/mL and the results in real samples expressed as mg/g, or what is the dilution factor for expressing μg/mL into mg/g. All these comments were ignored. How is possible that the results and discussion sections elaborate about twigs’ samples that are not acknowledged anywhere in the experimental section.
I should insist that the article is lacking important experimental details. For example, 47.014 g of extract were obtained for leaves, what about twigs? Is the dilution volume for these extracts (leaves and twigs) the same? What kind of test was used to confirm that the 7 calibration curves were linear? The authors should be aware that r2 equal/close to 1 (as in their Table 1) is not equivalent to linearity. Reporting r2 with 4 significant figures is no guarantee of linearity. The literature is full of mathematical models with r2<1 (e.g. 0.7-0.8) and with a high degree of linearity.
It is difficult to figure out the relation between table 1 (in µg/mL) and 2 table 2 (in mg/g).
One more time the authors stated (line 235) that the results are analyzed using a p-value of 0.05. However, a wide variety of p-values are reported throughout the manuscript (e.g. p<0.01 or 0.001 or 0.05). An explanation about the statistical grounds for comparing results at different p-values should be provided in the text.
Although Figure 2 was improved, the y-axes are reported without units.
A rough view of the reference section reveals 18% of errors do not comply with the guidelines of Nutrients. This is unacceptable for a revised version.
The authors stated in their rebuttal letter that all the references were revised in accordance with the guidelines of the journal. However, I can see that they only corrected those references that I mentioned in my first report and did not pay attention to the rest. In addition, it is preposterous that 57% of the new included references (4 out of 7) contain different kinds of errors. I suggest correcting references 6, 8, 18, 23, 25, 26, 33. And keep in mind that I spotted mistakes in these particular references by performing a rough view of the section. It is the authors duty to check the accuracy of “ALL” the references and also verify that they comply with the format suggested by Nutrients.
